# A flexible topo-optical sensing technology with ultra-high contrast

Cong Wang [1], Ding Wang[1], Valery Kozhevnikov[2], Xingyi Dai[3], Graeme Turnbull [2], Xue Chen[1], Jie Kong[3✉], Ben Zhong Tang [4✉], Yifan Li [1✉] & Ben Bin Xu [1✉]

Elastic folding, a phenomenon widely existing in nature, has attracted great interests to understand the math and physical science behind the topological transition on surface, thus can be used to create frontier engineering solutions. Here, we propose a topo-optical sensing strategy with ultra-high contrast by programming surface folds on targeted area with a thin optical indicator layer. A robust and precise signal generation can be achieved under mechanical compressive strains (>0.4). This approach bridges the gap in current mechano-responsive luminescence mechanism, by utilizing the unwanted oxygen quenching effect of Iridium-III (Ir-III) fluorophores to enable an ultra-high contrast signal. Moreover, this technology hosts a rich set of attractive features such as high strain sensing, encoded logic function, direct visualisation and good adaptivity to the local curvature, from which we hope it will enable new opportunities for designing next generation flexible/wearable devices.

[1] Department of Mechanical and Construction Engineering, Faculty of Engineering and Environment, Northumbria University, Newcastle upon Tyne NE1 8ST, UK. [2] Department of Applied Sciences, Faculty of Health and Life Science, Northumbria University, Newcastle upon Tyne NE1 8ST, UK. [3] MOE Key Laboratory of Material Physics and Chemistry under Extraordinary, Shaanxi Key Laboratory of Macromolecular Science and Technology, School of Chemistry and Chemical Engineering, Northwestern Polytechnical University, Xi'an 710072, China. [4] Department of Chemistry, The Hong Kong Branch of Chinese National Engineering Research Center for Tissue Restoration and Reconstruction and Institute for Advanced Study, The Hong Kong University of Science and Technology, Clear Water Bay, Kowloon, Hong Kong, China. ✉email: kongjie@nwpu.edu.cn; tangbenz@ust.hk; yifan.li@northumbria.ac.uk; ben.xu@northumbria.ac.uk

One of the latest trends in next generation micro-electronics technology is to develop flexible optical sensors and actuators, which holds promises in strain/pressure sensing[1–4], wearable devices[5–7], electronic skin[8–10], camouflaging[11], etc. By utilizing soft materials, recent efforts have explored the flexible optical technology with extra controllability and on-demand color changing such as triboelectric–photonic[12,13], piezo-electroluminescent[14], piezo-photonics[15–17], mechano-responsive luminescence (MRL) and mechanochromism[18]. Among those approaches, MRL, a tunable and switchable luminescence (or chromism) in response to mechanical stimulus[19,20], have attracted considerable interests for their potentials in sensing/micro-devices[21], data storage[22,23], flexible display[24,25], security pattern/inks[26], etc. However, the optical performance has been discounted by aggregation-caused quenching (ACQ)[27,28], thus limit the further applications for MRL materials. Whereas the current advances in Aggregation-induced emission (AIE) have achieved emergence characteristics at molecular level to overcome the drawbacks of ACQ[29–31], novel optical sensing mechanisms remain yet to be exploited to enable wider scale-up perspectives.

Inspired by epidermal color changing scheme from nature, researchers have been able to amplify signal by generating luminescent molecular dominos[19] thus realize multi-state optical switching by engineering micro/nano-structures on surface[18]. By far, all practiced strategies will easily result into a noisy and low-resolution signal, which poses challenges in triggering controllable signals for scalable applications. Subject to the mechanical stimuli, elastomeric materials can undergo surface morphological change (e.g. wrinkles and cracks) which has been used to create switchable optical features[25,32], and structural colour with dynamic luminescent patterns[33]. Zeng et al reported an interesting mechanochromic device by using cracks and folds[34] to trigger optical signals within a moderate stretching strain of 0.2. While the understandings on controllably generating elastic instability morphologies have been extended[35–38], even to form 3D structures[39–41], surface topology enabled optical sensing in response to large compressive strain (more than 0.4) has not been reported elsewhere.

In this work, we propose a topo-optical sensing strategy with ultra-high contrast by constructing a patterned elastic multilayer coated with a nanometer thin optical indicator layer. The keys to achieve such high contrast topo-sensing strategy include the targeted folding on elastic surface guided by the pre-defined lattice pattern and the autonomous optimization of contrast by selectively oxygen quenching of the coated Iridium-III complexes (Ir-III) fluorophore layer. The unique self-contact geometry of folding area preserves intensity by mechanically creating a hypoxia zone, whereas the intensity reduces significantly for the rest of surface due to the oxygen-quenching at the open air. Moreover, we successfully demonstrate several conceptual designs based on this topo-sensing approach such as an in-plane strain sensor, a 2D spy barcode, an adaptive topo-optical grid with potential for bio-applications and a flexible bending sensor, to shed the lights on the future applications in micro-devices and flexible/wearable electronics.

## Results

### Configuration of targeted folding on elastic multilayer.

The multilayer system consists of a soft polydimethylsiloxane film (PDMS, shear modulus $G_{sub} \approx 0.15$ MPa, thickness of 125 μm) on a vinylpolysiloxane mounting substrate (VPS, shear modulus $\approx 0.35$ MPa, thickness of 1.5 mm). Oxygen plasma was applied to create a hard skin layer (shear modulus $G_f \approx 1.8$ MPa, thickness of 100 nm) on the top of PDMS film. By applying a uniaxial compression (Fig. 1a), $\varepsilon_{comp} = \frac{L_0}{L} - 1$, an elastic morphological

development is shown on the surface. A similar setting has been previously used to configure wrinkle pattern by pre-placing Bravais lattice holes on the surface at low compression[42], where an unexpected formation of wrinkle-to-crease/folding transition was discovered occasionally under a higher compression ($\varepsilon > 0.4$) but have not been studied further. The key in this work is to investigate the controllable formation of targeted crease/fold at higher compression and translate this topographical transition into a dedicated sensing signal in responding to a compressive strain.

We first compare the development of elastic morphologies with reflective optical microscopy between a plain (Fig. 1b) and a patterned surface with a single-line array of micro-holes as shown in Fig. 1c (diameter = 60 μm, distance $D = 120$ μm, hole depth $h = 12$ μm, Supplementary Movie 1). Wrinkle patterns are developed globally for both plain and centre lattice hole patterned surfaces at low compressive strains and evolved into visible textures when strain increases to $\varepsilon = 0.27$. A strain energy localization guided by the pre-placed pattern can be clearly identified along the micro-holes array. The surface presents a post-wrinkling development with mixed morphologies at middle compressive strains (i.e. $\varepsilon = 0.38$). According to Kim and co-workers, surface wrinkles will first undergo period doubling, followed by the formation of creases under a modulus ratio ($G_f/G_{sub}$) between 5.86 and 13.89[43,44]. A threshold strain ($\varepsilon_{th}$) is the compressive strain when the first fold occurs on surface, which is variable against the setting factors for multilayer. Here, a compressive strain of $\varepsilon = 0.52$, which is slightly higher than $\varepsilon_{th}$, is chosen to compare folding conditions at the same energy level. We find that a few random folds (pointed by red arrows in Fig. 1b) appear on the plain surface, while a single big fold locates at the area that is defined by the pre-placed holes on surface (dotted line in Fig. 1c). Single-line array with varied pattern shapes (circles, diamonds, squares, triangles and hexagons, Supplementary Fig. 1) and different $D/\Phi$ (Supplementary Fig. 2) are also attempted, where a range of $\varepsilon_{th}$ from 0.42 to 0.58 can be achieved by designing the shape and $D/\Phi$. However, the $\varepsilon_{th}$ shows less sensitivity on the depth of lattice pattern (Supplementary Fig. 3), which agrees with the reported results on configuring the wrinkle patterns with Bravais lattice[45].

### Realization of topo-optical sensing.

The concept of translating surface topology into optical signal (Topo-optical sensing) is initially facilitated (Fig. 1a) through casting and drying a drop of solution containing 1.3 mM fluorescein o-acrylate (FoA) on the elastic surface, to fulfil the photo-luminescence function. When the fold occurs, the in-plane length ($L$) locally develops into a self-contact depth ($H_c$, Fig. 1a), lead to an optical signal from the top view because of the volumetric accumulation of intensity. By assessing the optical properties for the morphologies at ($\varepsilon = 0.52$) for both plain and micropatterned surfaces under laser scanning confocal microscopy (LSCM, Fig. 1d, e), a single-line optical signal is clearly shown on the location defined by the micro-pattern with an enhanced intensity (side view). We define a signal-to-noise ratio (SNR) as, $SNR = \frac{peak\ intensity}{noise}$, to quantitatively analyse the optical signal, where the peak intensity is collected from the signal of folded line and the noise represent the average luminescence signals originated from the background surface (excluding the folding line). In Fig. 1f, a higher SNR value is obtained for the patterned surface than that of plain surface. With a hole array, surface energy can be guided to form a single fold with a deeper self-contact (Fig. 1e), rather than a distractive energy localization with multiple folds/creases on the plain surface. From the LSCM 3D reconstruction image (green opened book, Fig. 1g) for the FoA patterned surface, we note that the

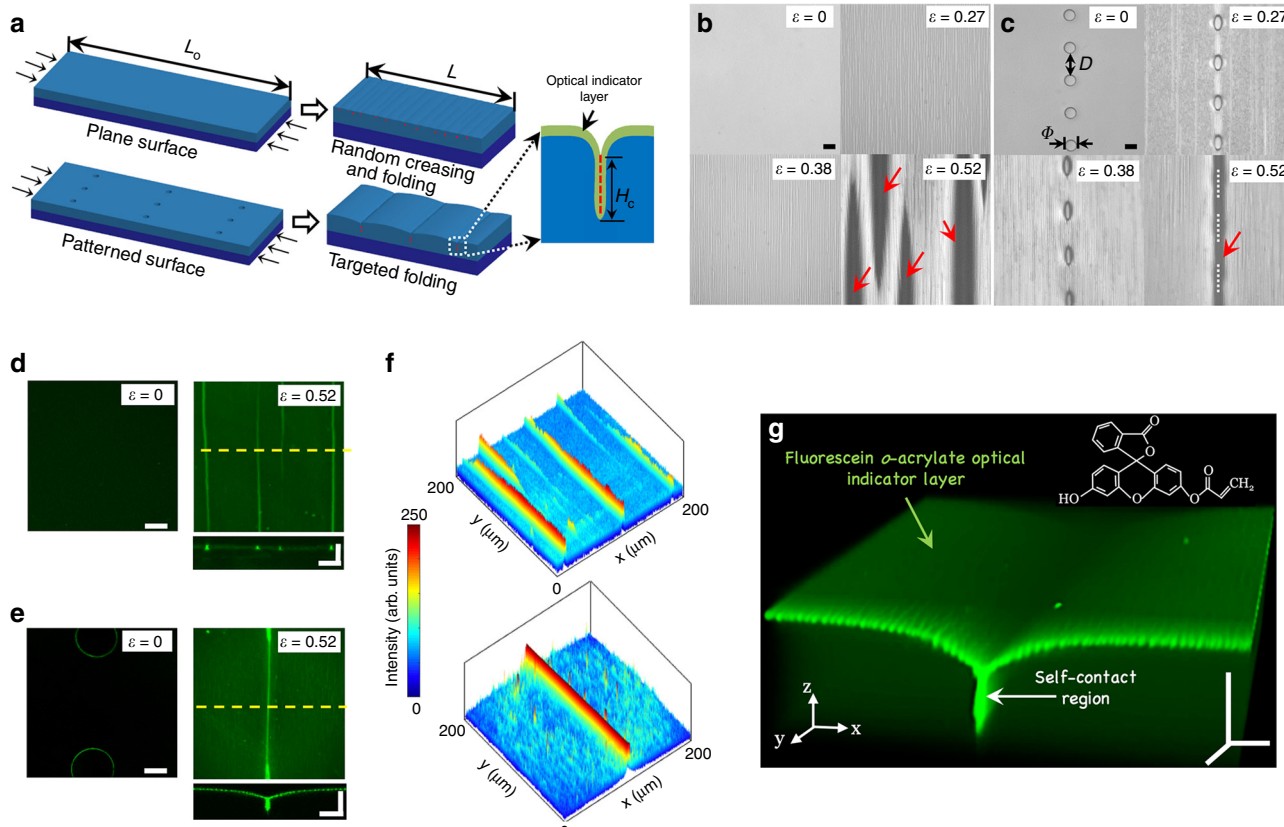

**Fig. 1 The realization of topo-optical sensing. a** Schematic illustrations and optical microscopic observations on the morphological changes on (**b**) plain surface and (**c**) surface patterned with single-line of micro-hole array ($\Phi = 60\,\mu m$, $D = 120\,\mu m$, $h = 12\,\mu m$, dot lines showing the location of closed holes) under compression. Laser scanning confocal microscopy (LSCM) images showing top and cross-sectional (for the dash lines in top view images) views of signals induced by (**d**) random folding on the plain surface, and (**e**) targeted folding on a micropatterned surface, with the digital analysed results for top view optical signals in (**f**). **g** A signal image of green opened book from 3D LSCM reconstruction to show the surface signal with targeted folding. All scale bars are 30 μm.

signal noise on background remains considerably high, due to the discontinuous fluorescein aggregation caused by small creases/folds[45].

**Selectively oxygen-quenching induced ultra-high contrast**. The Iridium-III (Ir-III) complex is an oxygen-quenching phosphorescent material which usually emits orange-red coloured light ($\lambda_{emission} = 580\,nm$, Fig. 2a) in hypoxia condition after being excited[45]. Our aim is to utilize the topological hypoxia zone, created by targeted folding, to preserve the optical signal on the self-contact region for the Ir-III coating layer (Fig. 2b). Exposed to the oxygen in open air, Ir-III complex luminescence outside of the folding area is mostly eliminated by the oxygen quenching effect, leading to a topo-optical signal with ultra-high contrast (see "blade" pattern in Fig. 2c and intensity analysis in Fig. 2d). A nominal line contrast (NLC) is defined as $NLC = \frac{peak\ intensity}{mean\ average\ of\ the\ line\ intensity}$, to describe the optical signal distribution for the selected area. After analysing the NLC data (Fig. 2e) for the selected lines in Figs. 2b, 1e, an NLC value of 10 is achieved for the Ir-III coated surface, which is 5 folds of the NLC (~2) for FoA coated surface.

We next scale this topo-optical relationship (SNR versus $H_c$, Fig. 2f) to understand the geometrical influence on the quality of signal. Small SNRs of $0.47 \pm 0.04$ are captured when the fold first occurs at $\varepsilon_{th}$ with an onsite $H_c \approx 1.1\,\mu m$, for both Ir-III and FoA coated surfaces. When $H_c$ grows higher than $1.6\,\mu m$, a stable SNR plateau (SNR ≥ 2) is emerging for FoA coated surface which

indicate that the physical accumulation along the folding depth reaches a threshold of intensity to enable a quality optical feedback. This development of self-contact depth, around 500 nm in distance, is very rapid within a strain window of ~0.037 ± 0.017. Given by a nominal strain speed of $0.02\,s^{-1}$, the sensing signal can be instantly captured in microscope within 1 s. In contrast, an increasing trend is obtained for Ir-III coated surface at the same threshold when $H_c$ increases, due to the oxygen quenching effect at the open air. When the $H_c$ reaches $13.8\,\mu m$, the SNR on Ir-III coated surface increases significantly to 12.5 which is 6 folds of that from FoA coated surface. By preserving the peak intensity at self-contact area, the Ir-III coated surface achieve higher SNRs when $H_c$ is larger than $1.6\,\mu m$.

The time-dependent degradation of photoluminescent signal is assessed by tracking the peak and background signals at $\varepsilon_{comp} = 0.55$ ($H_c \sim 13.8\,\mu m$) for up to 200 h. The results for FoA coated surface (Fig. 2g) show a retainment of intensity after 200 h with less than 8% decay for both peak and background signals. For Ir-III coated surface, a rapid decrease of around 92% intensity is discovered in first 6 h for the background signal, whilst the peak signal remains stable for the first 70 h, then starts to fade and reaches a decrease of 54% in 200 h. The quenching kinetics is analysed for the Ir-III films on PDMS surfaces with varied thicknesses (Supplementary Fig. 4) in the open air, where the optical signals are quenched for about 5 h in all layers without compression. Under compression, the topology enabled hysteresis results (Fig. 2g) into a high optical contrast for about 65 h, then the peak intensity eventually reduces due to the diffusion of

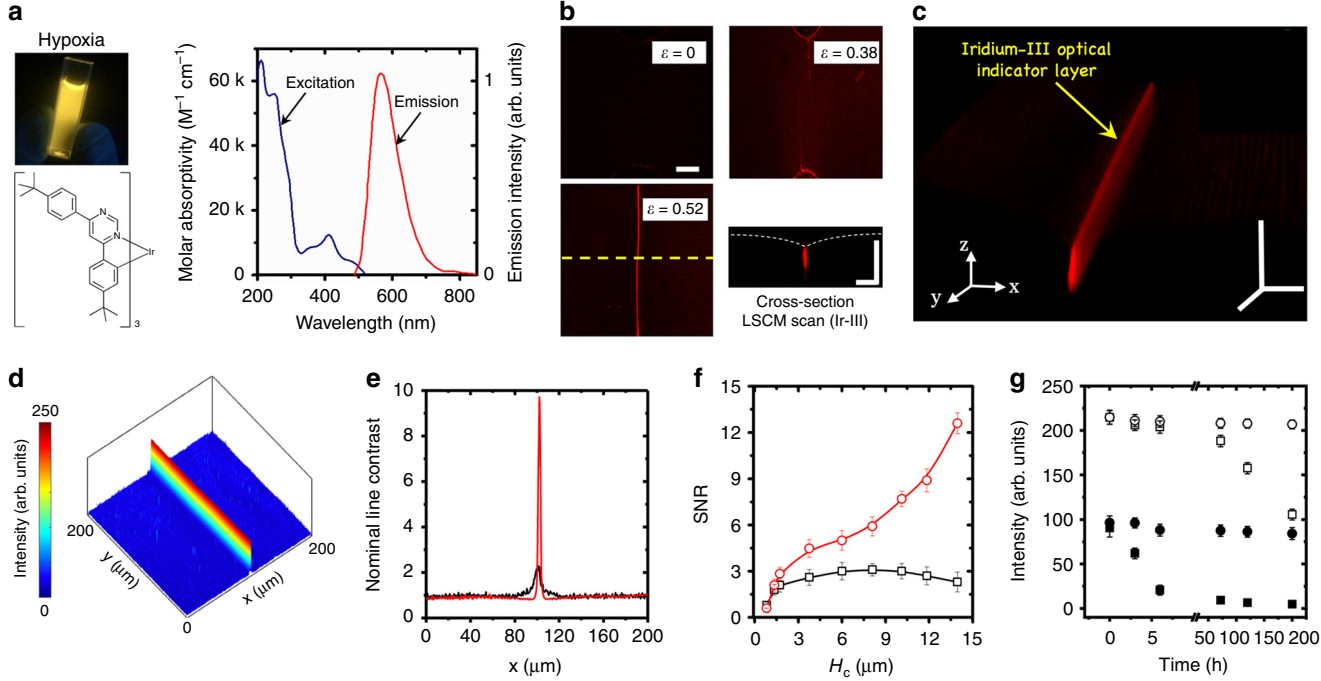

**Fig. 2 The generation of ultra-high contrast signal. a** Ir-III complex and its excitation and emission spectrum (**b**) Top and cross-sectional views of signal generated by an Ir-III indicator layer; (**c**) A signal image of red blade with ultra-high contrast signal from 3D LSCM reconstruction. **d** The analysed results for top view signal in the selected image in (**b**). **e** Comparison of the nominal line contrast (NLC) for the selected areas in (**b**) (red line) and Fig. 1e (green line). **f** The scaling of signal noise ratio (SNR) versus self-contact depth ($H_c$) for fluorescein o-acrylate (FoA, black symbol) and Iridium-III (IR-III, red symbol). **g** A time lapsing tracking of signal intensities in (**b**) and Fig. 1e at $\varepsilon_{comp}$ = 0.52, circle for FoA, square for Ir-III, hollow for the peak signals, solid for the background signals. All scale bars are 30 μm.

oxygen into the elastic solids. The kinetics of diffusing oxygen into the self-contact region of PDMS surface is complicated as it corresponds to the factors such as temperature, local oxygen concentration, humidity, surface porosity, chemical composition, etc, we thus defer this understanding into future study.

**Numerical analysis of self-contact depth guided by surface pattern.** We next perform numerical analysis with a commercial finite element simulation software to understand the mechanism of generating targeted folding. The single array of holes with varied geometrical inputs are considered to simulate the in-plane (Supplementary Movie 2) and out of plane (cross-section, Supplementary Movie 3) strain energy localization. By comparing the results for $D/\Phi = 1$ (Fig. 3a) and $D/\Phi = 5$ (Fig. 3b), we find that the in-plane strain concentration for $D/\Phi = 1$ is stronger than that for $D/\Phi = 5$. The out-of-plane (cross-section) simulation results suggest a progressing deformation with the closure of hole (initiation of $H_c$), development of $H_c$ as a folded contact with non-contact at the bottom (see Fig. 3b), then finally reaching a fully self-contact stage (creasing type, Fig. 3a).

As described above, the development of self-contact is rapid from an onsite $H_c$ at $\varepsilon_{th}$, to the $H_c$ that can provide enough contrast. We next numerically analysis the $\varepsilon_{th}$ as a function of $D/\Phi$ ($\Phi = 40$ μm) to study the threshold of generating optical signal guided by hole pattern. After comparing with the experimental results (Fig. 3c), the experimental results seem larger than the simulation results for $\Phi = 40$ μm, but good agreements on the overall trend are obtained for the surface, even for those surfaces patterned with different $D/\Phi$. It should be noted that we slightly over-compressed the surface to determine the closure stage for each hole under reflective optical microscope, due to the visco-elastic nature of materials. Thus, the

experimental $\varepsilon_{th}$ in this paper are a little larger than the simulated ones. The simulation for surface patterned with sharp corners (diamonds, squares, triangles and hexagons) are less successful at the moment as the current simulation programme does not allow the mesh process to progress over the sharp corner, we then separate the investigation in future work.

Following to the onsite of folding, further transient development of $H_c$ is critical in determining the intensity of optical signal. We plot nominal self-contact depths ($H_c/h$) for different $D/\Phi$ ($\Phi = 40$ μm) as a function of compressive strain (Fig. 3d) to understand this geometrical development. Excellent alignments are found between the numerical outcomes and experimental results for $D/\Phi = 1$ and $D/\Phi = 5$, meanwhile a slight mismatch is presented for $D/\Phi = 2$. A region (grey zone) with $H_c/h$ values between 0.15 and 1.2 located on a strain range of 0.39–0.59 is outlined by analysing the experimental results where we can expect the ideal optical signals. The results from cyclic testing (Fig. 3e) indicate a good resilience on generating target folding with a desired self-contact $H_c$ ($H_c$ of 10 μm for $\Phi = 40$ μm and a $H_c$ of 18 μm for $\Phi = 80$ μm, $D/\Phi = 2$), to create enough intensity of optical signal, after a short saturation period of 1–2 cycles. The marathon cyclic assessment reveals a reliable reproducibility of $H_c$ even after 100 cycles (Supplementary Fig. 5), due to the elastic nature of multilayer. Further study on the relaxation of folding was performed by tracing $H_c$ over a longer time duration, to compare with the simulation results (Supplementary Fig. 6a) at a nominal compressive strain ($\varepsilon_{comp}$) of 0.5. The result indicates a limited relaxation in $H_c$, which is in the same trend with the theoretical approximation. Low hysteresis is observed during the compression/recovery cycle (Supplementary Fig. 6b). The reason could be the elastic nature of multilayer system and low surface tension[45,46] after being coated by Ir-III compound. We then extend the surface design to the square lattice patterns

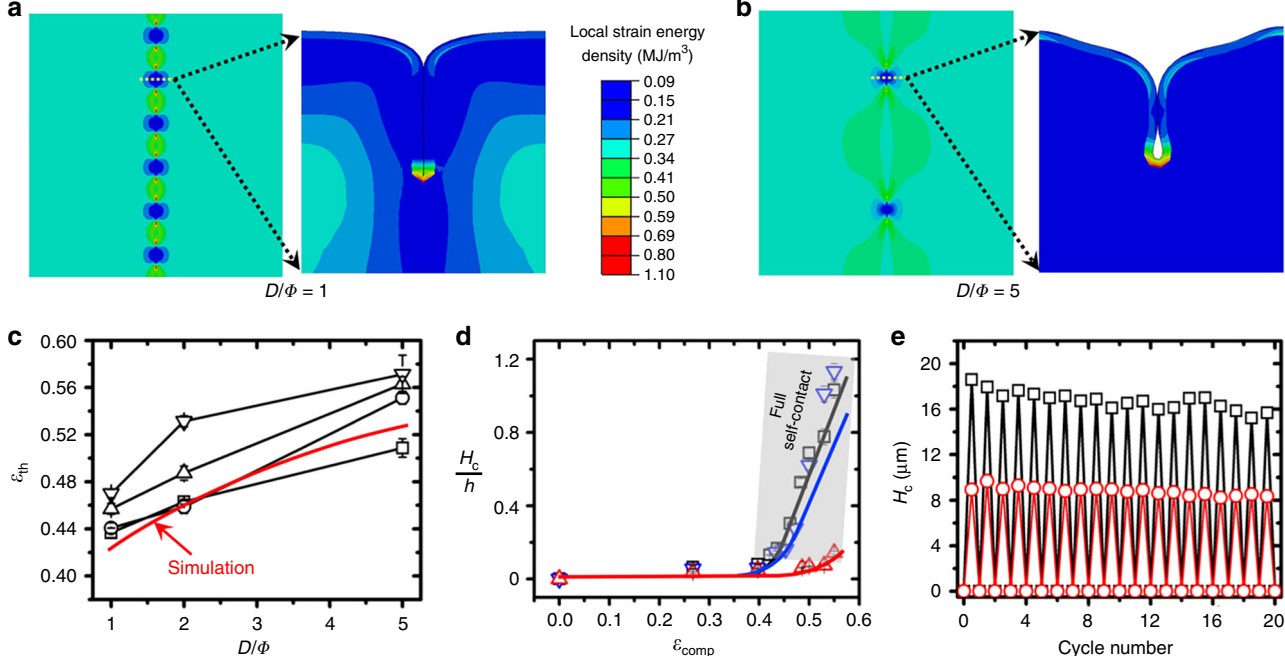

**Fig. 3 Numerical analysis and reproducibility assessment.** In-plane and out-of-plane strain energy analyses for the surfaces with a single micro-hole array ($\Phi = 40$ µm) of (**a**) $D/\Phi = 1$ and (**b**) $D/\Phi = 5$, at $\varepsilon_{comp} = 0.5$. **c** Comparison of the simulated $\varepsilon_{th}$ (threshold strain of hole closure) with experimental ones at different $D/\Phi$ for the surface with hole array ($\square$ for $\Phi = 20$ µm, O for $\Phi = 40$ µm, $\Delta$ for $\Phi = 60$ µm, $\nabla$ for $\Phi = 80$ µm). **d** Comparison the simulation (line) and experimental (symbol) results for nominal self-contact depth ($H_c/h$) under a progressive $\varepsilon_{comp}$ for the surface with a single micro-hole array ($\Phi = 40$ µm; black for $D/\Phi = 1$, blue for $D/\Phi = 2$, red for $D/\Phi = 5$). **e** The cyclic testing results with $\Phi = 80$ µm ($\square$) and $\Phi = 40$ µm (O), at $D/\Phi = 2$.

(Supplementary Movie 4), where $W$ is defined as the distance between neighbouring lines (Supplementary Fig. 7). The results suggest that varied $\varepsilon_{th}$ at high strain can be achieved by designing $D/\Phi$, $W/D$ of the lattice pattern, together with more capabilities on 2D designs.

**Demonstration of potential applications**. To demonstrate the potential of developing this topo-optical sensing mechanism into device applications, an in-plane topo-optical sensor (Fig. 4a) is presented to detect large surface strains, by simply configuring the pattern parameters (shape, $D/\Phi$, etc) for the pre-placed lattice (Supplementary Movie 5). A programmable stepwise switching mechanism is encoded in this design, where a reversible line pattern could be logically switched between '0,0,0' at $\varepsilon = 0$, '1,0,1' at $\varepsilon = 0.44$ and '1,1,1' at $\varepsilon = 0.52$, with corresponding optical signals can be visualized by reflective optical microscopy and fluorescence microscopy at the same time. This concept can be further developed into dynamic 2D spy barcode products with hidden information only appearing under a dedicated stain (Fig. 4b) and an adaptive topo-optical luminescence grid (Fig. 4c), which contains a tuneable feature on the size of grid under equi-biaxial compression to track the cell behaviour.

Based on this topo-optical sensing strategy, a flexible bending sensor can be developed by combining the in-plane pre-compression ($\varepsilon_{pre-c}$, Fig. 4d) to detect out-of-the-plane bending degree. After releasing pre-stretching strain of substrate (stage I), we deploy the device on the area to detect the bending level. By observing under microscopy, the device first experiences a selective fold on the lattice patterned surface at low degree bending (stage II), then all lattice patterns are folded at high degree bending (stage III). A brief phase diagram is created to distinguish the two-stage bending sensing for the patterned surface ($\Phi = 80$ µm, $h = 12$ µm, $D/\Phi = 1$ (black line) and 5 (red line), $W/\Phi = 5$), where a clear map is obtained to determine localized curvature with the provided $\varepsilon_{pre-c}$ when the optical signal occurs.

## Discussion

We have described a topo-optical sensing strategy by targeted generating folds on a micropatterned surface, with a coated optical indicator layer. The elastic multilayer shows a robust and precise optical signal by activating folds at the pre-patterned area under certain strain values. The formation of folding patterns has been studied with various geometrical inputs of the lattice patterns and the results are in a good agreement with the predictions from numerical analysis. An inherited automatic optimization on optical contrast is also introduced by oxygen quenching the Ir-III based optical indicator layer, which lead to an ultra-high contrast by significantly reducing the background noise. We anticipate this high-contrast topo-optical sensing strategy with the demonstrated conceptual devices will open new windows for future applications such as flexible/wearable electronics and bio-devices.

## Methods

**Fabrication of patterned multilayer elastomeric substrate**. The lattice patterns were prepared through SU-8 pillar array templates photo-lithographically patterned on silicon wafer (Supplementary Fig. 8) and followed by a stamp transfer (Supplementary Fig. 9). Single-line and square (multi-line) lattice arrays of SU-8 pillars were lithographically patterned on silicon substrates to create the stamp template. Firstly, 1, 1, 1, 3, 3, 3-hexamethylsilazane (ACROS ORGANICS) was spin-coated (30 s, 1000 rpm) onto the silicon wafer to promote adhesion. A thin layer of SU-8 (2025, Micro Chem) was then spin-coated, followed by soft baking at 95 °C for 5 min, before being exposed to UV light under a mask aligner (EVG 620). Post-exposure-bake was then performed (65 °C for 1 min, then ramped to 95 °C for 5 min). After being developed in an EC (ethylene lactate based) solution for 5 min, the patterned SU-8 templates were cleaned by isopropyl alcohol and DI (de-ionized) water. It was baked for another 15 min at 200 °C before stamp transfer.

The mounting substrates were made from vinylpolysiloxane elastomer (VPS, Elite Double 22, Zhermack) and cut into rectangular strip (9 mm wide, 30 mm long and 1.5 mm thick). The VPS strips were then mounted on mechanical vices (Supplementary Fig. 9), before being pre-stretched uniaxially to 600% strain. A thin layer (~125 µm) of polydimethylsiloxane (PDMS, Sylgard 184, base-to-crosslinker weight ratio = 30:1) was spin-coated on the fabricated SU-8 pillars, followed by 60 min soft bake at 70 °C. An adhesion PDMS layer was then spin-coated (30:1) onto the soft-baked PDMS to bond it to the mounting VPS layer. The multilayer structures were cured at 70 °C for 8 h. An air plasma treatment (100 w, Henniker

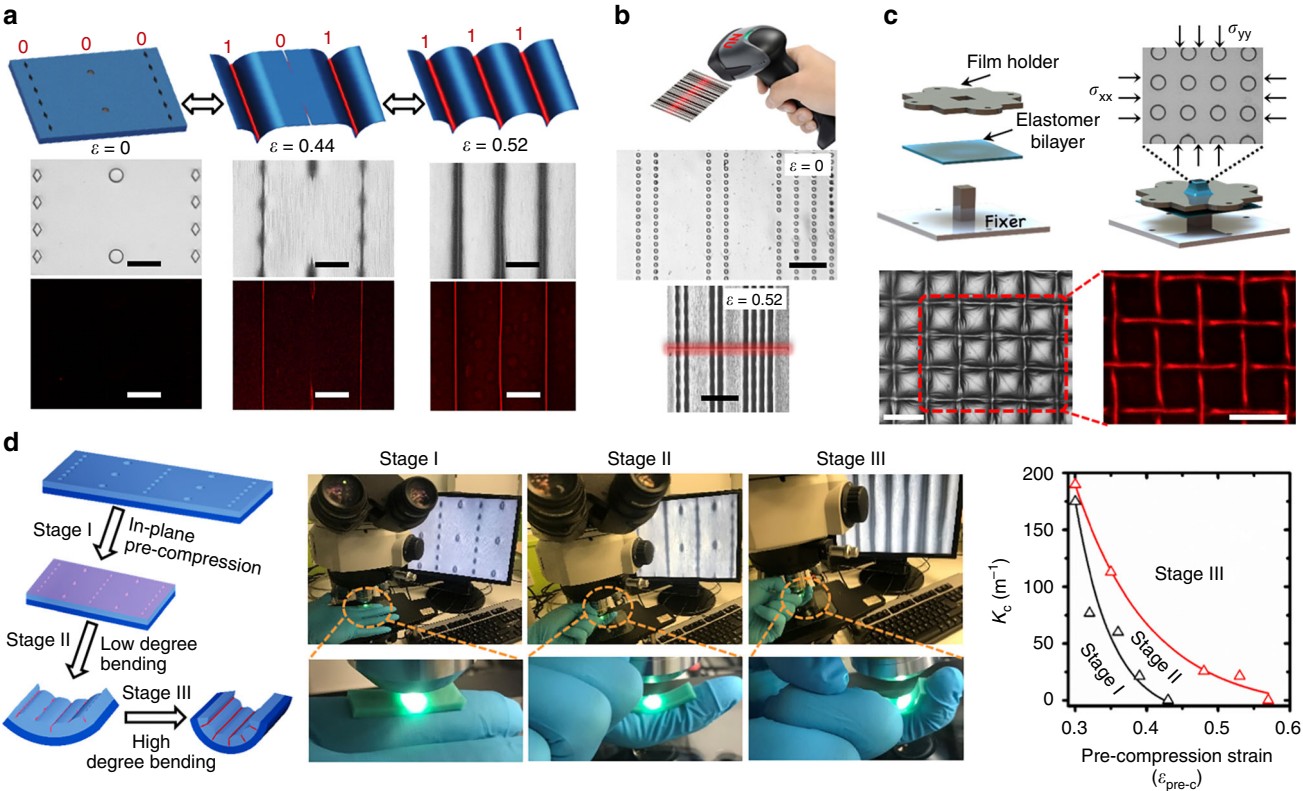

**Fig. 4 Demonstration of application concepts. a** An in-plane logic strain sensor with stepwise responses for large deformation sensing. **b** A 2D spy barcode design with a state of 'OFF' at $\varepsilon = 0$ and an 'ON' state at $\varepsilon = 0.52$. **c** An adaptive topo-optical grid under equi-biaxial compression. **d** A flexible tactile sensor to detect the change on local curvature with (left) schematic illustration of sensing principle, (middle) observation of conformal sensing under reflective microscopy, (right) phase diagram of the conformal tactile sensing. All scale bars are 100 μm.

HPT-100) of 10 s was applied to create a thin hard skin layer on the lattice patterned PDMS surface.

**Solution casting of optical indicator layer**. After the air plasma treatment, optical indicator solutions (1.5 mM) were prepared by dissolving the dye into the absolute ethanol and chloroform. A droplet of solution containing either fluorescein O-acrylate (Sigma-Aldrich) or Iridium-III complex (synthesized in house following the published route[47]) was then casted on the surface of multilayer. The solution droplet then spread and dried at room temperature to form an optical indicator layer with a measured thickness of ~600 nm.

**Characterizations**. Upon releasing the pre-stretched VPS mounting substrates, the PDMS thin layer experienced uniaxial compression. Incremental deformation in a unit nominal strain of ≈0.004 was applied to the sample during the compression (progressing) or tension (withdrawing/recovery) by a fixed amount at regular intervals in room temperature. A reflective optical microscope (Nikon LV-100) was used for observation under white light. 3D and 2D fluorescence imaging was performed using Nikon A1R LSCM system (LSCM). For all observations/tests, multiple measurements were performed on 7–15 selected samples (areas) to minimize the system error. For the fluorescein O-acrylate images, the scanner selection was set to be Galvano, with laser excitation wavelength of 488 nm and emission wavelength of 540 nm. For the Iridium-III images, the scanner selection was set to be Galvano, with laser excitation wavelength of 406.6 nm and emission wavelength of 595 nm. The captured fluorophore images were subsequently analysed by the MATLAB to get its light intensity data and image (surf, shading interp). Standard deviations (error bars in the figures) were calculated based on the mean averaging of a group of data from 7 to 9 independent measurements on different samples.

**Numerical simulation**. We used the commercial simulation package—ABAQUS, to simulate surface folding on the multilayer under uniaxial compression. The incompressible neo-Hookean material model was used for all elastic materials in this analysis. Structural symmetry was assumed when the fold is simulated. The pseudo-dynamic method incorporated in ABAQUS was adopted. The geometrical inputs have been magnified by 1000 times due to the limitation of mesh size in ABAQUS. An element type CAX8H was used for mesh.

## Data availability

The data that support the findings of this study are available via Northumbria Research Data Management scheme and per request from the corresponding author (B.X.).

## Code availability

The numerical code developed in this work is available upon request from Dr. Xue Chen (sherry.chen@northumbria.ac.uk).

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

## Acknowledgements

The work was supported by the Engineering and Physical Sciences Research Council (EPSRC) grant-EP/N007921/1 and EP/S01280X/1, and Royal Society Kan Tong Po International Fellowship 2019-KTP/R1\191012. Dr Ben Xu and Dr Ding Wang would thank Reece Innovation for the studentship support. Professor Jie Kong thanks the financial support from Natural Science Basic Research Plan in Shaanxi Province (2018JC-008, Distinguished Young Scholar).

## Author contributions

The idea was initially generated between B.X., Y.L. and V.K. B.X. and Y.L. directed the research programme and designed the experiments with C.W. C.W. carried out the major part of experiments with D.W.'s assistances on multilayer fabrication and pattern transformation. The micro-pattern template was engineered by D.W. V.K. and G.T. synthesis and characterized the Ir-III complex and diluted it into the mixed solvent for casting. B.X., X.C. and X.D. developed the analytical simulations. B.X., Y.L. J.K. V.K. and B.Z.T analyzed and interpreted the data. B.X., Y.L. J.K. and B.Z.T defined the scope together and wrote the paper with contributions from all authors.

## Competing interests

The authors declare no competing interests.
