## [Peer Review File · Nature Communications]

Reviewers' comments:

Reviewer #1 (Remarks to the Author):

The authors of "A Flexible Topo-optical Sensing Technology with Ultra-high Contrast" describe an approach to create a high contrast topo-sensing system using a patterned multi-layer elastomer coated with a thin optical indicator layer. This approach relies on targeted folding of the surface in order to optimize the signal contrast by selectively quenching the thin optical indicator layer. The self-contacting regions of the crease preserves the intensity of the signal while the intensity of the signal is reduce for the rest of the surface due to oxygen-quenching of the Iridium-III complexes fluorophore (Ir-III). While targeted folding of the surface using the Bravais lattice is not new and has been previously reported on by the authors (Wang, et al, Adv. Funct. Mater. 28, 1-9, 2018), the optimization of the signal using this system is. I believe this paper will be of interest to others in the community as well as influence thinking of the field especially with the section of the paper which demonstrated several applications of this topo-sensing approach. In addition, the authors have provided enough detail for a researcher to reproduce their work. I recommend this paper to be published in Nature Comm. with some minor revisions:

1. Add scale bars to Figure 1g, Figure 2c and Figure 4c
2. Typo in line 228, should be analyze and not "analysis"
3. Several typos in the reference section

Reviewer #2 (Remarks to the Author):

The authors have studied an interesting phenomenon of luminescence of elastic polymer surface under high strain folding, and have demonstrated its potential applications. The paper is well written.

Although in Figure 3e change of H_c with a few applied cycles are shown, the authors should discuss more on the durability issue since the material is under high strain compression and question of how it might sustain many repeated load cycles for it to be practically applicable should be addressed.

I recommend the paper be published in Nature Communications with minor revision.

Reviewer #3 (Remarks to the Author):

The author used mechanical force induced surface folding topographies as a platform for optical indicator. The Iridium-III (Ir-III) fluorophores with oxygen quenching effect are applied as the source for fluorescent signal. The closed structure of the surface folding generated by bending allows this fluorophore to demonstrate bright fluorescence. Thus, a new type of structure-oriented mechanoluminescent materials was present. The idea itself is of interesting, but the authors need to address to the following concerns before considering for publication.

- 1) The time lapsing of the signal intensity start to drop at 75 h for the Ir-III intensity at $\epsilon = 0.52$ as shown in Figure 2(g) probably due to the oxygen permeation. This might not be promising for strain sensing for a long period of time (such as for several weeks), the reviewer wonder if there's any method to improve the time dependent decay of the signal intensity?
- 2) The stress/strain relaxation of PDMS can change the local strain energy density of the folding structure over long time. This might also be a possible factor contribute to the time dependent decay of the signal intensity. Can the author demonstrate any experimental result or simulations for how this stress/strain relaxation behavior of the folding strain energy density over long time?
- 3) The author should present the result how fast the fluorescent signal can be response to the mechanical force. The response speed is crucial for sensing some dynamic system, like human finger motion.
- 4) The author only demonstrates the mechanoluminescence can be seen under fluorescent microscopy which can limit this system to be applied for wider application field. Thus, the author should present whether this mechanoluminescent response can be seen clearly with high contrast under macro-scale (like center-meter scale), which, ideally, can be captured by the eyeball directly and clearly.

Point-by-point response to reviewer's comments

=====

Reviewer #1

=====

The authors of "A Flexible Topo-optical Sensing Technology with Ultra-high Contrast" describe an approach to create a high contrast topo-sensing system using a patterned multi-layer elastomer coated with a thin optical indicator layer. This approach relies on targeted folding of the surface in order to optimize the signal contrast by selectively quenching the thin optical indicator layer. The self-contacting regions of the crease preserves the intensity of the signal while the intensity of the signal is reduce for the rest of the surface due to oxygen-quenching of the Iridium-III complexes fluorophore (Ir-III). While targeted folding of the surface using the Bravais lattice is not new and has been previously reported on by the authors (Wang, et al, *Adv. Funct. Mater.* 28, 1-9, 2018), the optimization of the signal using this system is. I believe this paper will be of interest to others in the community as well as influence thinking of the field especially with the section of the paper which demonstrated several applications of this topo-sensing approach. In addition, the authors have provided enough detail for a researcher to reproduce their work. I recommend this paper to be published in *Nature Comm.* with some minor revisions:

Reply from Authors:

We thank this reviewer for the comments and suggestions. While we appreciate the reviewer for his/her interests on our researches, we slightly disagree on the comments of '*folding of the surface using the Bravais lattice is not new and has been previously reported on by the authors (Wang, et al, Adv. Funct. Mater. 28, 1-9, 2018)*'.

Our previous research (*Adv. Funct. Mater.* 28, 1-9, 2018), as quoted by the reviewer, mainly investigated the formation and configuration of lateral wrinkle patterns with the dependencies on the geometrical variables of in-plane confinements. The studies were mostly based on the elastic surface patterned with a different type of lattice array (Fig 1b in <https://onlinelibrary.wiley.com/doi/pdf/10.1002/adfm.201704228>). The undertaken experimental observations and simulations in previous AFM paper mostly focused on relatively low compressive strains (i.e. $\epsilon < 0.35$), due to the mechanics's nature of wrinkle initiation and relative post-wrinkle development (please see p. 5 of 9 in <https://onlinelibrary.wiley.com/doi/pdf/10.1002/adfm.201704228>). The transition from wrinkling to creasing at high compressive strain values described in AFM paper, provided indicative data with the schematic drawings and the threshold strains captured when the fold occurs, but no systematic studies (experimental or theoretical) on the morphologies and the associated developments.

From the perspective of mechanics, we believe the study on the controllable formation of targeted crease/fold at higher compression (i.e. $\varepsilon > 0.4$) in this research is new and can be distinguished substantially from the research shown in our AFM paper. Relative statement has been highlighted in the second paragraph at page 4 (p.4) in the revised manuscript.

1. Add scale bars to Figure 1g, Figure 2c and Figure 4c

Reply from Authors:

Scale Bars have been added in the above figures as advised.

2. Typo in line 228, should be analyze and not “analysis”

Reply from Authors:

This typo has been corrected.

3. Several typos in the reference section

Reply from Authors:

These typos have been corrected, along with a number of others identified during further proofreading of the manuscript.

=====
Reviewer #2
=====

The authors have studied an interesting phenomenon of luminescence of elastic polymer surface

under high strain folding, and have demonstrated its potential applications. The paper is well written. Although in Figure 3e change of H_c with a few applied cycles are shown, the authors should discuss more on the durability issue since the material is under high strain compression and question of how it might sustain many repeated load cycles for it to be practically applicable should be addressed.

I recommend the paper be published in Nature Communications with minor revision.

Reply from Authors:

We thank the reviewer for his/her assessment of our manuscript and constructive suggestions. The generation of instability morphologies (i.e. wrinkle, creases, folds, etc) have been proved as highly reversible and robust (Phy. Rev. Lett. 109, 038001, 2012; Adv. Mater. 26, 4381, 2014; Angew. Chem. Int. Ed. 56, 6523, 2017; JMPS 123, 305, 2019), due to the elastic nature of multi-layer. Figure 3e intended to show the saturation process of this reversible self-contacting which represented the self-adaption of the multi-layer system to reach the desired H_c for sufficient contrast, as well as the durability for the initial 20 cycles. However, we agree with the reviewer that the testing results should be given at considerable amount of cycles, to validate the durability of system under high compressive strains for more cycles. We hereby add the following data as new Fig S5 in the SI file and describe this trend in the second paragraph in p.10.

Figure S5 The cyclic testing results (up to 100 cycles) for samples with $\Phi = 80 \mu\text{m}$ and $\Phi = 40 \mu\text{m}$ ($D/\Phi=2$).

=====
Reviewer #3
=====

The author used mechanical force induced surface folding topographies as a platform for optical indicator. The Iridium-III (Ir-III) fluorophores with oxygen quenching effect are applied as the source for fluorescent signal. The closed structure of the surface folding generated by bending allows this fluorophore to demonstrate bright fluorescence. Thus, a new type of structure-oriented mechanoluminescent materials was present. The idea itself is of interesting, but the authors need to address to the following concerns before considering for publication.

Reply from Authors:

We thank the reviewer for carefully reading the manuscript and considering our research as 'new' and our idea as 'interesting'. The points raised by reviewer are important to improve the manuscript.

1) The time lapsing of the signal intensity start to drop at 75 h for the Ir-III intensity at $\epsilon = 0.52$ as shown in Figure 2(g) probably due to the oxygen permeation. This might not be promising for strain sensing for a long period of time (such as for several weeks), the reviewer wonder if there's any method to improve the time dependent decay of the signal intensity?

Reply from Authors:

Our aim was to investigate the controllable formation of targeted crease/fold at high compression ($\epsilon > 0.4$) and translate this topographical transition into a dedicated sensing signal. The disruptiveness of this work also creates an inherit optical self-optimisation mechanism by selectively oxygen quenching the coated Ir-III based optical indicator layer, therefore yield an ultra-high contrast by significantly reducing the background noise.

We note that, unfortunately, the absolute intensity in self contact region will drop after certain period due to the oxygen sensitivity nature of Ir-III, the relative study has been described in the second paragraph in p.8 (Fig S4). However, a high contrast window of ~ 70 hours should be feasible for some applications such as logic signal coding/decoding and security bar for hidden code scanning/reading (\sim seconds, Figure 4a&b). The potential application for adaptive topo-optical grid could be promising for some biological applications (minutes to days), where the oxygen concentration can be controlled at relatively low level in bio-media (i.e. aqueous solutions). Therefore, the high contrast window would be likely to be extended to weeks or even longer. As pointed by the

reviewer, the development of long lasting/permanent high contrast topo-optical sensing mechanism is of great importance, which we have been discussing with our partner who specialized in luminescent materials and planning for the next stage study.

2) The stress/strain relaxation of PDMS can change the local strain energy density of the folding

structure over long time. This might also be a possible factor contribute to the time dependent decay of the signal intensity. Can the author demonstrate any experimental result or simulations

for how this stress/strain relaxation behavior of the folding strain energy density over long time?

Reply from Authors:

While the studies on nucleation and growth of creases/folds in the PDMS based elastic multi-layer usually took it as a pure elastic system, there have been questions on the possible relaxation/hysteresis on the generation of creases/folds during cyclic testing and/or compression-recovery curve, and the dependencies on the system settings. Previously, Suo, Heyward, et al presented an interesting study (Phy. Rev. Lett. 109, 038001, 2012) indicating that the surface energy is the major factor to influence the relaxation/hysteresis of the crease/fold - *'nucleation and growth of crease are resisted by the surface energy Adhesion, rather than plastic deformation, is responsible for the dramatic hysteresis'* To echo this comment, we add new supplement data as new Figure S6 and describe them in p.11 in the revised manuscript with new ref 45 &46.

Figure S6 (a). Comparison the simulation and experimental results for relaxation behaviour under a progressive $\epsilon_{comp} = 0.5$ for the surface with a single micro-hole array ($\Phi = 40 \mu\text{m}$, $D/\Phi = 5$). (b) The hysteresis results for targeted folding depth on the surface with a single micro-hole array ($\Phi = 40 \mu\text{m}$, $D/\Phi = 1$ and $D/\Phi = 5$).

3) The author should present the result how fast the fluorescent signal can be response to the mechanical force. The response speed is crucial for sensing some dynamic system, like human finger motion.

Reply from Authors:

Thanks for this important advice. We apologize that this carelessness for missing critical data. We now added the *'Given by a nominal strain speed of 0.02 s^{-1} , the sensing signal can be instantly captured in microscope within 1 second.'* in the first paragraph in p.8, accompanied by the updated statement of *'Incremental deformation in a unit nominal strain of ≈ 0.004 was applied to the sample during the compression (progressing) or tension (withdrawing) by a fixed amount at regular intervals in room temperature.'* in the section of *'Characterizations'* in p.14.

4) The author only demonstrates the mechanoluminescence can be seen under fluorescent microscopy which can limit this system to be applied for wider application field. Thus, the

author should present whether this mechanoluminescent response can be seen clearly with high contrast under macro-scale (like center-meter scale), which, ideally, can be captured by the eyeball directly and clearly.

Reply from Authors:

The Iridium-III (Ir-III) complex typically emits orange-red coloured light ($\lambda_{\text{emission}} = 580 \text{ nm}$, Figure 2a) in hypoxia condition after being excited, which falls within the visible range of 390-700 nm for human eye. Thus, the signal should be able to be seen by the naked human eye in theory. However, the self-contact depth we actuated, H_c , was normally in the range from a few micron metres to tens micro metres. Given by casting a thin optical indicator layer in nanometres on the top, the signal is subtle that can only be identified by the human eye conditionally, i.e. with the assistant from instrument. Therefore, most optical performances in this research were quantified by microscope. In addition to the potential opportunity to be integrability into micro-electronics device, the described indirect mechanism also inspired an idea of spy coding of hidden information (Fig 4b) which requires a conditional reader/kit to translate the message.

In terms of the scaling-up demonstration at macroscopic level, we did achieve a visibly identifiable morphology for folding line with a length of $\sim 10 \text{ mm}$ as shown in the following figure R1. It is under reflective light and doesn't require extra luminescent response. This part was not considered as mandatory part as we considered it slightly deviated from the scope of disruptiveness in this paper - *topological transformation enabled optical sensing with ultra-high contrast signal*. Indeed, the suggested idea of macroscopic Topo-optical sensing is interesting for its convenient implementation/integration in scaling-up applications, which we include it as part of our future work.

Figure R1 The observation of actuated fold for the surface with a single micro-hole array ($\Phi = 40$ μm , $D/\Phi=2$) at macroscopic level under reflective light.

REVIEWERS' COMMENTS:

Reviewer #2 (Remarks to the Author):

The authors have addressed all queries satisfactorily. I recommend it be published in Nature Communications.

Reviewer #3 (Remarks to the Author):

The author already properly address to my comments. It can be considered to published in the present form.